# Nanoscale size effects in α-FAPbI₃ evinced by large-scale ab initio simulations

Virginia Carnevali [1,3], Lorenzo Agosta[1,3], Vladislav Slama [1], Nikolaos Lempesis [1,2], Andrea Vezzosi[1] & Ursula Rothlisberger [1] ✉

Formamidinium-lead-iodide (FAPbI₃) has a rich phase diagram, and long-range correlation between the organic cations and lattice dipoles can influence phase transitions and optoelectronic properties. System size effects are crucial for an appropriate theoretical description of FAPbI₃. We perform a systematic ab initio study on the structural and electronic properties of the photoactive phase of FAPbI₃ as a function of system size. To ensure an accurate theoretical description, three criteria must be satisfied: the (correct) value of the band gap, the extent (or the absence of) structural distortions, and the zeroing out of the total dipole moment. The net dipole moment vanishes as the system size increases due to PbI₆ octahedra distortions rather than due to FA⁺ rotations. Additionally, thermal band gap fluctuations are predominantly correlated with octahedral tilting. The optimal agreement between simulation results and experimental properties for FAPbI₃ is only achieved by system sizes approaching the nanoscale.

Metal halide perovskites (MHP) are one of the most promising classes for the photoactive layer (AL) of photovoltaic (PV) materials[1,2]. Thanks to their remarkable optoelectronic properties, such as a high absorption coefficient, tunable band gap, high charge carrier mobility, and low exciton binding energy, perovskite-based solar cells (PSCs) lead to high photo conversion efficiencies (PCE) and solar cell performance. Perovskites have the chemical formula ABX₃, where, in the case of MHPs, A is an organic or inorganic cation, B is a divalent metal cation, and X is an anion of the halogen group. Efficient PSCs require uniform and defect-free perovskite thin films over large areas, which can improve charge transport, suppress non-radiative energy loss, and minimize device degradation pathways[3,4]. So far, among all MHPs, formamidinum-lead-iodide (FAPbI₃) in the cubic symmetry phase (α-FAPbI₃) yields the best PV performance. The reported experimental band gaps for this phase are in the range of 1.45–1.51 eV[2,5–7], which is close to the ideal single-junction Shockley-Queisser band gap of 1.31 eV[8]. Besides, compared with other MHPs such as methylammonium lead iodide (MAPbI₃), FAPbI₃ exhibits an improved thermal stability due to higher activation energies for thermal degradation[9]. These two factors help to establish FAPbI₃ as

the most promising AL among the MHPs for single-junction PSCs with a PCE record of 26.7%[10].

However, the pursuit of enhanced stability and performance in FAPbI₃ has prompted experimentalists to seek theoretical support to elucidate the pivotal mechanisms involved in surface passivation and charge transport, which are critical to guide the design of new PSCs[11–14]. In response to this need, theorists have developed surface and bulk models of FAPbI₃ with an appropriate level of accuracy that can mimic the electronic and structural features of FAPbI₃ films. Nevertheless, numerous other salient issues persist in their theoretical intricacy. These include phenomena such as nucleation from solution, crystal growth, halide segregation in mixed halide MHPs, and defect formation, which still need to be resolved at the atomistic level. To address these collective mechanisms, simulations are required that employ an adequate level of accuracy while permitting a substantial increase in the number of atoms. Because of the system size requirements, the type of simulation techniques used to simulate FAPbI₃ phases have been mostly in the framework of classical molecular dynamics (CMD) and Monte Carlo (MC) simulations based on empirical force fields[15–18], or density functional tight binding approaches[17,18]. In this regard, also

[1]Laboratory of Computational Chemistry and Biochemistry, Institute of Chemical Sciences and Engineering, Swiss Federal Institute of Technology (EPFL), Lausanne, Switzerland. [2]Present address: Department of Chemistry, University of Ioannina, Ioannina, Greece. [3]These authors contributed equally: Virginia Carnevali, Lorenzo Agosta. ✉e-mail: ursula.roethlisberger@epfl.ch

force-matched force fields and machine-learning potentials trained on ab initio molecular dynamics (AIMD) data are promising solutions because they allow CMD to be performed with AIMD accuracy, provided the physics is correctly described in the training data set[19–21]. For the generation of sufficiently accurate reference data, it is evident that an adequate and appropriate description of $FAPbI_3$ at the quantum level is of paramount importance.

In this context, there is an important amount of computational literature on $FAPbI_3$ - and MHPs in general - where different periodic models and levels of theory are used, sometimes leading to contradictory results for fairly fundamental properties such as band structure and band gap[22–24], suggesting a lack of predictive capability. Several computational methods have been adopted to determine the band gap of $FAPbI_3$, spanning from density functional theory (DFT) with different generalized gradient approximations[25,26], meta functionals or (range-separated) (meta) hybrid functionals[27], van der Waals functionals[28,29], to the GW approximation of many-body perturbation theory, or combinations of them[30,31]. In addition, spin-orbit coupling (SOC) is usually implemented in the presence of Pb/Sn atoms, which leads to a significant lowering of the band gap[25,32]. In the context of lead perovskites, the combination with hybrid functionals has been empirically demonstrated to effectively mitigate the impact of SOC on the band gap[33]. A fairly comprehensive summary of the band gap calculated for $FAPbI_3$, depending on the level of theory used, is provided in this review[34]. Apart from the employed theoretical method, the band gap can also be influenced by the choice of system size. In fact, part of the variations in the computed $FAPbI_3$ properties with different levels of theories might also be due to different choices of system sizes, and in particular, the use of small supercells, which imposes an ordered molecular orientation of ferroelectric multidomains, which is not observed experimentally[35–38]. There are some examples in the literature in which larger system sizes are adopted, obtaining interesting results. Carignano et al.[39] utilize fairly large $FAPbI_3$ supercells to analyze the cation dynamics with AIMD. As demonstrated by Ummadisingu et al.[38] and Ma et al.[40], the electronic structure of $MAPbI_3$ must be studied at the nanoscale to ensure an accurate capture of its properties. This was recently confirmed experimentally by Weadock and coworkers[41]. A similar approach was adopted by Wiktor et al.[42] for $CsMX_3$ (M = Sn, Pb; X = Cl, Br, I). Evidence has also been found for electron-phonon coupling in $FAPbI_3$ as a cause of long-range optical phonon modes, indicating large size effects[43]. Using CMD, Maheshwari et al.[44] demonstrated that phase transitions in hybrid halide perovskites are driven by a complex interplay between dipole–dipole interactions between organic cations and the metal halide lattice, resulting in the formation of large, organized domains of organic cations. This has significant implications for the electronic structures of these materials.

In general, system size plays a crucial role in the accuracy and predictive power of DFT simulations of perovskite materials, its importance varying significantly for different classes of perovskites. In organic-inorganic hybrid perovskites such as $FAPbI_3$ and $MAPbI_3$, the dynamic behavior of the organic A-site cations and the soft, anharmonic lattice framework introduce complex structural fluctuations and symmetry-breaking distortions that are not well captured by small simulation cells[39,41]. The same is expected for the Sn counterparts, $FASnI_3$ and $MASnI_3$, but with a less pronounced system size effect due to the stiffer structures resulting from a more covalent Sn-I bond compared to the more ionic Pb-I bond[45]. For example, the orientation and rotation of the $FA^+$ or $MA^+$ cations affect hydrogen bonding with the halides, local electric fields, and ultimately the band gap, dielectric screening, and polaron formation[38,40]. In contrast, inorganic halide perovskites such as $CsMX_3$ (M = Sn, Pb; X = Cl, Br, I) exhibit reduced orientational disorder due to the isotropic nature of the $Cs^+$ ion but still require moderately large supercells ($\sim 2 \times 2 \times 2$ or larger) to accommodate the octahedral tilting patterns and electron-phonon coupling effects that influence their optoelectronic behavior[43]. Meanwhile,

oxide perovskites such as $BaTiO_3$ are relatively rigid and structurally less complex, allowing ferroelectric behavior and structural distortions to be reliably captured even in smaller supercells[46,47]. However, even for oxide perovskites, larger cells may be needed to model phase transitions or strain effects[48]. Overall, while system size affects all perovskite simulations to some extent, the need for large, thermally sampled simulation cells is particularly pronounced in soft hybrid halide systems, where neglecting size and disorder effects leads to significant underestimation of band gap, carrier dynamics, and structural stability.

Despite experimental and computational insights that point to a dependence of the electronic band gaps of MHPs and the system size, a direct assessment of this effect is missing. Computer simulations in the context of ab initio molecular dynamics would be able to tackle the complexity of this challenge and help to explain the experiments by linking electronic structure and size effects. However, the limitations imposed in length and time scales by ab initio calculations discouraged the exploration of large systems in the field of MHPs, favoring simulations with small supercells with a high level of theory, as previously mentioned.

In this work, we want to address the key questions: what are the system size and level of theory needed to properly capture the structural and electronic properties of $FAPbI_3$ and is the static zero Kelvin description sufficient or are simulations at finite temperature required? These are crucial questions from both theoretical and experimental perspectives. Using AIMD at 300 K and first-principles calculations, we characterize the structural and electronic properties of $\alpha$-$FAPbI_3$ with increasing size of the simulation cell at the DFT level with both the PBE and PBE0 functionals. SOC is also considered when allowed by memory and/or computational resources. Our simulations show that it is necessary to use sufficiently large simulation cells (at least a 768-atom cell) in which all degrees of freedom (atomic positions and simulation cell) are fully relaxed in order to obtain a non-distorted $\alpha$-$FAPbI_3$ structure, a converged band gap and a small overall dipole moment. For 0 K calculations, it is also essential to set up an initial system configuration in which the FAs cations are randomly oriented according to the 3-fold symmetry (Supplementary Fig. 1). Furthermore, with large simulation cells (from 2592 atoms upwards), PBE is able to describe the electronic band gap in good agreement with the experimental value when calculated as thermal average over finite-temperature equilibrated AIMD snapshots. We also show that the net dipole moment of the system goes to zero as the cell size increases, and this is related to long-range effects due to the $PbI_6$ octahedral tilting. Finally, we demonstrate that thermal band gap oscillations are mainly related to the octahedral tilting.

## Results

### Characterization of α-FAPbI₃ at 0 K

The obtained band gaps for different theoretical methods and computational schemes are reported in Table 1 as a function of system size, ranging from the minimal primitive cell (12-atoms) to larger supercells containing increasing even numbers of primitive cells. The periodicity imposed by the size of the supercell has to be compatible with the periodicity in the octahedral tilting[49], while the 12-atom cell that is not compatible with this condition (but is still an often-used setup in $FAPbI_3$ calculations) was included as a reference. At 0 K, we ran two series of calculations using the initial structure $\alpha$-$FAPbI_3$ from the material project database[50]. In the first case (relax), only the atomic positions were optimized, keeping the simulation cell fixed to cubic, while in the second (vc-relax), both the atomic positions and the lattice parameters were optimized. We also set up two initial configurations: (i) all–aligned: FA molecules are kept aligned with their dipole moments pointing all in the same direction in order to have a meaningful comparison between the 12-atom cell and all the other

## Table 1 | Band gap

| Simulation cell | NPT-F PBE 300 K (eV) | NPT-F PBEO 300 K (eV) | relax PBE 0 K (eV) | vc-relax PBE 0 K (eV) | vc-relax PBE + SOC 0 K (eV) | vc-relax PBEO 0 K (eV) | vc-relax PBEO + SOC 0 K (eV) | $n \times n \times n$ k-point grid |
|---|---|---|---|---|---|---|---|---|
| 12-atom (1 × 1 × 1) | 3.62 ± 0.21 | 5.08 ± 0.21 | 3.68 | 3.72 | 3.46 | 5.82 | 5.54 | 1 |
| | - | - | 2.21 | 2.31 | 1.78 | 3.90 | 3.33 | 2 |
| | - | - | 1.48 | 1.97 | 1.25 | 3.47 | 2.72 | 4 |
| | - | - | 1.49 | 1.63 | 0.49 | 3.44 | 2.28 | 6 |
| | - | - | 1.49 | 1.56 | 0.45 | 3.38 | 2.25 | 8 |
| | - | - | 1.48 | 1.55 | 0.45 | 3.37 | 2.24 | 10 |
| 96-atom (2 × 2 × 2) | 2.16 ± 0.17 | 2.80 ± 0.15 | 2.24 | 2.26 | 1.78 | 3.36 | 2.85 | 1 |
| | - | - | 1.58 | 1.69 | 0.60 | 2.84 | 1.63 | 2 |
| | - | - | 1.49 | 1.58 | 0.46 | 2.73 | (1.61) | 4 |
| | - | - | 1.48 | 1.61 | 0.54 | 2.75 | (1.68) | 6 |
| | - | - | 1.61* | 1.60* | 0.48* | 2.76* | (1.64)* | 6 |
| 768-atom (4 × 4 × 4) | 1.76 ± 0.04 | 2.28 ± 0.03 | 1.54 | 1.60 | 0.47 | - | - | 1 |
| | - | - | 1.74* | 1.74* | 0.63* | - | - | 1 |
| | - | - | 1.48 | 1.56 | 0.44 | - | - | 2 |
| | - | - | 1.69* | 1.69* | 0.56* | - | - | 2 |
| 2592-atom (6 × 6 × 6) | 1.59 ± 0.07 | 2.17 ± 0.03 | 1.51 | 1.71 | - | - | - | 1 |
| | - | - | 1.50* | 1.50* | - | - | - | 1 |
| 6144-atom (8 × 8 × 8) | 1.47 ± 0.08 | 2.09 ± 0.05 | 1.47 | 1.67 | - | - | - | 1 |
| | - | - | 1.68* | 1.69* | - | - | - | 1 |

FAPbI$_3$ Kohn-Sham band gap for different system sizes at several theory levels and system optimization schemes. The initial configurations for the 0 K simulations were chosen with all FAs aligned except the ones indicated with a star superscript. AIMD was performed at 300 K in NPT-F, and the band gaps were computed as averages along the trajectories (4–7 ps) after equilibration. Values in brackets are additive estimates from SOC and PBEO calculations since direct calculations require too much memory for available resources. Source data are provided as SourceData_Table1.txt.
*FA pseudo-randomly oriented.

## Table 2 | Structural parameters and dipole moment

| Simulation cell | MSE 0 K | $\theta$ 0 K(deg) | $|d_{Tot}|$ PBE 0 K (Debye/ABX$_3$) | $|d_{Tot}|$ PBE 300 K (Debye/ABX$_3$) | $|d_{Tot}|$ PBEO 300 K (Debye/ABX$_3$) |
|---|---|---|---|---|---|
| 12-atom (1 × 1 × 1) | 0.08 | 1.37 | 13.17 | 2.32 ± 0.98 | 1.80 ± 0.70 |
| 96-atom (2 × 2 × 2) | 0.78 | 18.03 | 3.36 | 2.24 ± 1.02 | 4.57 ± 0.71 |
| | 0.09* | 17.36* | 3.25* | | |
| 768-atoms(4 × 4 × 4) | 0.30 | 5.65 | 1.18 | 0.92 ± 0.26 | 0.89 ± 0.25 |
| | $4 \cdot 10^{-3}$* | 6.46* | 1.10* | | |
| 2592-atom (6 × 6 × 6) | 0.23 | 5.18 | 0.39 | 0.39 ± 0.12 | 0.31 ± 0.09 |
| | $8 \cdot 10^{-4}$* | 5.67* | 0.36* | | |
| 6144-atom (8 × 8 × 8) | 0.25 | 4.11 | 0.13 | 0.24 ± 0.07 | 0.24 ± 0.07 |
| | $1 \cdot 10^{-4}$* | 5.52* | 0.18* | | |

Mean squared error (MSE) of the lattice vectors with respect to the perfect cubic FAPbI$_3$ α-phase (column 1) and absolute maximum value of the octahedral tilting angle (column 2) at 0 K. For the structure optimized in the presence of all-aligned FAs, the octahedral tilting angle refers to the global deformation of the octahedra, resulting in a tilting angle of θ for all octahedra; while for the structure optimized from pseudo-randomly oriented FAs the tilting angle refers to the average absolute value of the typical–θ/θ tilting pattern (Supplementary Fig. 3). Total dipole moment ($|d_{Tot}|$) per stoichiometric unit (ABX$_3$) for different simulation cells; the dipole moment was computed at 0 K (column 3) and after equilibration as average along the AIMD trajectories performed at 300 K in the NPT-F ensemble (columns 4-5). The initial configurations for the 0 K simulations were chosen with all FAs aligned except the ones indicated with a star superscript. Source data are provided as SourceData_Table2.txt.
*FA pseudo-randomly oriented.

supercells; (ii) pseudo−random: FAs are pseudo-randomly oriented (from the 96-atom supercell upwards), where the overall orientation of FAs preserves the threefold symmetry resulting into a null total dipole moment of the system (Supplementary Fig. 1). In all cases, k-point sampling was increased until convergence was achieved.

For both the relax and vc-relax calculations, the band gap converges with increasing k-point grid but to different final values, indicating that both atomic positions and lattice parameters, i.e. lattice constants and angles (vc-relax) must be relaxed. For all the simulation cells with dipole-aligned FAs, the relax band gap converges to a value around 1.48 eV. At first glance, this is in close agreement with the experimentally measured value for the α-FAPbI$_3$ band gap and can be justified by the fact that, by keeping the simulation cell cubic, we are, in a sense, simulating the average (pseudocubic) α-FAPbI$_3$; however, with a closer look at the structure, we show that this is actually an artifact. Indeed, in addition to the correct band gap, it is necessary to verify that the cell distortions from the α-FAPbI$_3$ structure upon vc-relax and the total dipole moment of the supercell also have physical meaning. The vc-relax band gap of the 96-atom cell converges to a slightly higher value with respect to the 12-atom one. This is related to the significant distortion of the crystal structure from the cubic symmetry after vc-relax, decreasing the overlap of the electronic orbitals and consequently opening the band gap[45]. In principle, this finding aligns with experimental observations that indicate the orthorhombic perovskite phase as the most stable at low temperatures[51]. To quantify the deviations from cubic symmetry, we have analyzed the mean squared error (MSE) between the perfect α-FAPbI$_3$ structure and the vc-relax one, as well as the distribution of the octahedra tilting angles after vc-relax for the two cases—with and without pseudo-random FA orientation (Table 2). In the pseudo-random FA orientation case, the initial total dipole moment is smaller than $10^{-3}$ Debye. The pseudo-random orientation of the FAs in the starting configuration allows cubic symmetry to be maintained almost perfectly, as the MSE is reduced by two orders of magnitude for all supercells compared to the case with fully aligned FAs. The distributions of the octahedral tilting angles of the structures optimized from the pseudo-randomly oriented FA configurations average 0 degrees as expected (Supplementary Fig. 2), and the maximum value for the tilting angle converges with increasing

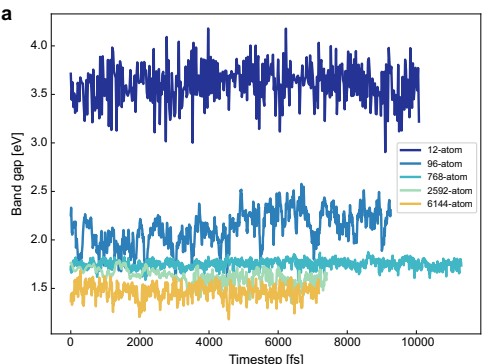
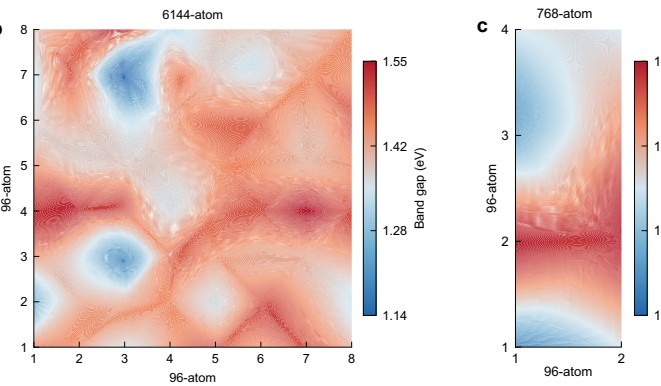

**Fig. 1 | Band gap oscillations and domains. a** Time variation of the band gap along the PBE NPT-F trajectory at 300 K for different supercells. Spatial band gap variation on 64 × 96-atom subcells contained in the 6144-atom FAPbI₃ supercell (**b**) and on 8 × 96-atom subcells contained in a 768-atom supercell (**c**). Source data are provided as a zip folder SourceData_Fig1.

system size (Table 2). The 96-atom cell has a wide spread distribution because of the significant lattice distortions after vc-relax. In contrast, the structures optimized with all-aligned FAs do not present an octahedral tilting pattern. Local distortions of the octahedra due to a collective upward or downward motion of I ions compensate for the strongly directional dipole due to the dipole-aligned FAs (Supplementary Fig. 3). A comparison of all-aligned and pseudo-random configurations reveals that the vc-relax band gap is only improved for the 2592-atom (6 × 6 × 6) cell. This phenomenon may be attributed to the fact that only this particular supercell allows for the complete relaxation of all degrees of freedom. This is because both the threefold symmetry for FAs and the octahedral tilting pattern are satisfied. It has been observed that, for all supercells, the potential energy with pseudo-randomly oriented FAs is consistently lower than that of the all-aligned case (Supplementary Fig. 4). This finding suggests that the pseudo-random configuration is a more favorable option for the system. Indeed, all-aligned configuration has a net total dipole moment for the FA that induces structural distortions in the Pb-I cages to compensate for the overall dipole moment that costs energy.

By adding SOC, the band gaps of the 12- and 96-atom cells converge to slightly different values −0.45eV and 0.54eV, again indicating a potential problem in the description of the electronic structure related to the cell distortions for the 96-atom cell. The band gap of the 768-atom cell with SOC, also converges to the same value as the 12-atom cell, and this may be related to the conservation of cubic symmetry by the 768-atom cell. Remarkable is what is obtained with the PBE0 functional. It is known that for materials such as lead halide perovskites, SOC and PBE0 contributions should cancel each other out[25]. The results of our 0 K calculations show that this effect only comes into play at k-point convergence for the 96-atom cell, whereas it is entirely absent for the 12-atom cell; by moving from the 12-atom cell to the 96-atom cell, the electron charge localization decreases as the band gap correction due to the inclusion of PBE0 decreases by 0.65 eV, allowing for SOC compensation and convergence to a band gap of 1.60 eV. It is evident that the size of the system plays a crucial role in reproducing the empirical findings of canceling out SOC-PBE0 contributions in the band gap and ensuring the structural properties of the system converge well. It was not feasible to run PBE0 + SOC for systems larger than 96-atom due to the excessive memory requirements of the simulations. However, making use of the observed cancellation between PBE0 and SOC effects also for the largest cells, the 2592-atom cell with a band gap of 1.50 eV (Table 1) recovers the experimental band gap. Even though PBE0 and SOC effects cancel each other out once the system size reaches convergence, thereby leaving the band gap unchanged with respect to the PBE vc-relax value, PBE0 + SOC

values offer a more accurate estimation of the absolute position of the valence band maximum (VBM) and conduction band minimum (CBM). This enhanced accuracy can be attributed to a reduction in self-interaction error and an improved description of the asymptotic behavior of the exchange-correlation potential, which is of paramount importance for the calculation of correct band alignments and the estimation of defect formation energies[52]. The symmetry of the band edge states also changes because of the band splitting induced by SOC. The (0 K) Kohn-Sham eigenvalues of the VBM and CBM obtained for fully optimized structures are reported in Suppl. Tab. 1. At the PBE0 + SOC level of theory, we obtain a lowering of ∼0.6 eV of the PBE Kohn-Sham eigenvalues even if the PBE0 and SOC effects cancel each other out for the band gap.

## Characterization of α-FAPbI₃ at 300 K

The thermally-averaged FAPbI₃ band gap was also calculated at 300 K by performing AIMD in the isothermal-isobaric ensemble with flexible cell option (NPT-F) at the PBE and PBE0 levels of theory for a minimum of 7 ps and up to 11 ps (Table 1). The band gap was computed as the average along the equilibrated AIMD trajectory (Fig. 1a). We run in NPT-F to avoid any kind of symmetry restriction to the system. Since we are operating at 300 K, the initial orientation of the FAs is no longer important, as the kinetic energy is sufficient to randomize the FA orientation after a few AIMD steps. The finite temperature PBE band gap for the 6144-atom cell is 1.47 ± 0.08 eV, which matches very well the experimentally measured gap, illustrating again the efficient compensation of many-body and SOC effects. Indeed, for all simulation cells, PBE0 (without SOC) consistently leads to a strong overestimation of the band gap.

We also exploit the larger system sizes studied by AIMD, together with its statistical approach, to study the instantaneous band gap dependence from the local lattice distortions. We have subdivided the 6144-atom and 768-atom cells into 64 and 8 96-atom cells, respectively, and computed the local band gap by projecting the density of states locally along the NPT-F trajectory (Fig. 1b, c). The 3D arrangement of the supercells was projected into a 2D map for better visualization (Supplementary Fig. 5). There is a local band gap variation for both supercells of about 1 eV, but in the 6144-atom cell there are more defined band gap domains, while for the 768-atom cell the band gap is more homogeneous throughout the supercell. This again indicates the importance of the system size in order to detect the possible formation of band gap domains. It further indicates that this is a phenomenon that has to be related to the process of long-range relaxation, stressing once again the importance of correctly describing the mechanisms that we have previously shown to be related to the size of

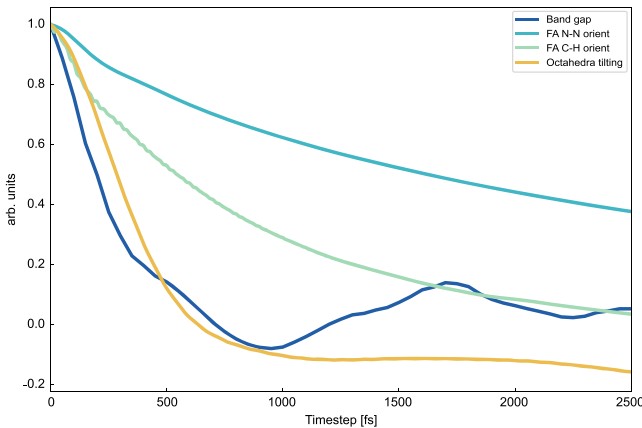

**Fig. 2 | Band gap correlation.** Time correlation function characterizing the rotational dynamics of FA cations along the N-N (blue curve) and C-H (orange curve) axes, as well as the octahedral tilting (red curve), and band gap oscillations (green curve). The analysis was done on the 6144-atom supercell over an equilibrated MD trajectory of 5 ps at 300 K. Source data are provided as zip folder SourceData_Fig2.

**Table 3 | FA dipole moment contribution versus FA starting configurations**

| FA orientation | $|d_{Tot}|$ (Debye/ABX$_3$) | $|d_{FA}|$ (Debye/ABX$_3$) | $|d_{Tot}-d_{FA}|$ (Debye/ABX$_3$) |
|---|---|---|---|
| all-aligned | 0.82 | 0.30 | 1.02 |
| random | 1.00 | 0.12 | 1.08 |
| random_best | 0.94 | 0.06 | 1.02 |
| smart_100 | 1.05 | 0.00 | 1.05 |
| smart_quasi | 1.16 | 0.00 | 1.16 |

Modulus of the total dipole moment of the system ($|d_{Tot}|$) separated into the modulus of the FA contribution ($|d_{FA}|$), and modulus difference between the dipole moment and the FA contribution to the total dipole moment of the system ($|d_{Tot}-d_{FA}|$). The values were computed for a 768-atom cell on equilibrated AIMD snapshots starting from an initial configuration with different FA orientations. The all-aligned, random and random_best, smart_100 and smart_quasi systems are initiated from an initial configuration of FAs that are all aligned, randomly oriented, and pseudo-randomly oriented, respectively (Supplementary Fig. 8). The random and smart configurations are built with a null initial total dipole moment. Source data are provided as SourceData_Table3.txt.

the system, such as the octahedral tilting[40,44]. To quantify the connection between band gap fluctuations and octahedral tilting, we have calculated the time correlation function of the band gap oscillations and the octahedral tilting for the 6144-atom cell (Fig. 2), getting correlation times of about 25 fs and 30 fs, respectively, i.e. very similar time scales suggesting that the band gap variations are indeed closely connected to changes in octahedral distortions. The influence of the tilting angles on the band gap directly aligns with established literature, as the tilting angles impact the antibonding overlap of the orbitals involved in the band edges[50]. Furthermore, the FA cations motion can be characterized by two characteristic rotational correlation times of 80 fs and 250 fs for the C-H and N-N vectors, respectively (Fig. 2), that are farer from the value calculated for the band gap oscillations. We have also computed the time correlation functions of the bottom of the conduction band and top of the valence band eigenvalues obtaining a trend comparable with that of the band gap (Supplementary Fig. 6). Local variations in the band gap might indicate that FAPbI$_3$ can potentially absorb photons with different wavelengths in different regions of the sample, which may be a further explanation of why this material performs as well as AL. The electronic charge distribution due to lattice motions in halide perovskites have implications on thermal fluctuations in the electronic structure. It has been shown that off-centering of Sn$^{2+}$ and Br$^-$ widens the band gap of CsSnBr$_3$[53]. Additionally, the top of the valence band and the bottom of the conduction band of FAPbI$_3$ are dominated by I$^-$ and Pb$^{2+}$ contributions, respectively[23]. For these reasons, we can expect that the fluctuations in the band gap occur on time scales similar to those characterizing the octahedral tilting fluctuations. The selection of an appropriate system size is imperative for the accurate description of the octahedral tilting pattern. Our findings demonstrate that PbI$_6$ tilting converges for a 768-atom supercell. The last point needed to verify the proper description of FAPbI$_3$ is the dipole moment.

### Dipole moment
The dipole moment was computed at 0 K and at 300 K (Table 2). In principle, the total dipole moment should be zero; however, for small-size simulation systems, it is not, due to the presence of residual, not fully compensated dipoles. Given the substantial decrease in dipole moment observed with increasing system size, it can be deduced that this phenomenon is associated with a long-range collective interaction, such as the long-range dipole-dipole correlations present between FA-FA and octahedral cage dipoles[44]. FA has a non-zero permanent dipole moment, while the octahedral distortions, i.e.,

contraction or elongation along different I-Pb-I axes, might induce a significant dipole moment. Table 3 reports the dipole moment contributions obtained by subtracting the dipole moment of the FA cations in the initial configuration from the total dipole moment of the system. The result is similar for all configurations (Table 3), and comparable with the values in Table 2, concluding that the initial FA configuration does not affect the overall dipole moment of the cell. To estimate the actual contributions of FA and octahedral cage distortions to the dipole moment, two NPT-F simulations were performed with FA respectively PbI$_6$ octahedra frozen. The octahedral distortion compensates for the dipole moment of the system in cases the FA cation orientations create a non-zero dipole moment (Supplementary Fig. 7). The time correlation function associated to the dipole moment fluctuations shows that the correlation time (~30 fs) is not affected by the system size (Supplementary Fig. 9) at contrast to the absolute value of the dipole moment (Table 2). Except for the 12-atom and 96-atom cells, which have already shown large size effects for structural as well as electronic properties, there is no major change in dipole moment when using PBE0. Finally, the values of the dipole moment obtained with PBE and PBE0 are statistically equivalent for the 6144-atom cell (Table 2). This finding suggests that the residual dipole moment may not be directly associated with the degree of charge localization imposed by the functional.

### Discussion
In conclusion, by means of large-scale first-principles calculations and ab initio molecular dynamics simulations at 300 K, we demonstrated that in order to get an accurate description of the structural and electronic properties of the α-phase of FAPbI$_3$, the size of the simulated system needs to approach the nanoscale. In particular, we showed that three conditions have to be met simultaneously, namely a proper description of the band gap, minimization of structural distortions, and the zeroing out of the total dipole moment. For first-principles calculations, it is essential to start from an initial configuration where the FAs are pseudo-randomly oriented by preserving the 3-fold symmetry and minimizing the dipole moment. At 300 K, because of the finite temperature dynamics, the initial configuration of the FAs is not stringent, and, from a 2592-atom cell upwards, the PBE approximation is already able to describe the electronic band gap of α-FAPbI$_3$. For the 6144-atom cell, we have computed a band gap of 1.47 ± 0.08 eV, which is in excellent agreement with the experimental values of 1.45–1.51 eV reported in literature (highlighting that PBE0 and SOC corrections only cancel out for this system size range). The evaluation of the absolute positions and symmetry of VBM and CBM with PBE will be affected by a significant self-interaction error and the wrong asymptotic behavior of the exchange-correlation potential.

**Table 4 | Partial charges are used for computing the classical dipole moment**

| Atom | Charge | Description |
|------|--------|-------------|
| C | −0.06 | - |
| N | −0.19 | - |
| $H_C$ | +0.27 | H bonded to C |
| $H_{NC}$ | +0.28 | H bonded to N closer to C |
| $H_{NN}$ | +0.30 | H bonded to N further from C |
| Pb | +2 | - |
| I | −1 | - |

Partial charges for Pb, I, C, N, and H atoms taken from our in-house point charge force field for FAPbI3. These charges were used for computing the classical dipole moment of the supercells.

These errors can be reduced by performing a single-point energy calculation on the PBE-optimized structures using PBE0 + SOC. Despite the fact that PBE0 and SOC effects cancel each other out for the band gap, as is often observed for lead halide perovskites, these corrections provide a more accurate estimation of the absolute values of the Kohn-Sham eigenvalues. A memory bottleneck prevented the completion of a convergence study on PBE0 + SOC for the larger supercells; however, an assessment of the convergence on the smaller cells permitted the confident assumption of mutual cancellation of PBE0 and SOC effects also on the larger supercells. Always, the 6144-atom cell minimizes structural distortions with respect to the perfect α-FAPbI3 structure and has the lowest dipole moment among all the systems studied. A significant correlation was discovered between $PbI_6$ octahedral tilting, band gap oscillations, and dipole moment. In particular, the dipole moment goes to zero only if the system size is large enough to properly relax the tilting pattern of the octahedra. Overall, an adequate size of the system (at least a 6144-atom cell) is needed to correctly describe its physics, as we have demonstrated with the identification of band gap domains related to a correct description of the octahedral tilting. Our work provides a detailed insight into the connection between structural and electronic properties of α-FAPbI3 – and MHPs in general – making an important contribution to the field of ab initio simulations dedicated to understanding fundamental physical principles, such as hole-electron transport, which is of paramount importance in the development of increasingly high-performance PSC devices.

## Methods First-principles calculations

DFT simulations at 0 K were performed with the Quantum ESPRESSO (QE)[54] suite of codes. All the calculations were run with DOJO fully relativistic norm-conserving PBE pseudopotentials[55,56] and well-converged basis sets corresponding to an energy cutoff of 150 Ry for the wave functions and 600 Ry for the charge density. Different k-point Monkhorst-Pack grids[57] were used, all centered on Γ-point. Semi-empirical corrections accounting for the van der Waals interactions were included with the DFT-D3 approach[58]. Different simulation cells were used, starting from a $1 \times 1 \times 1$ α-FAPbI3 (12-atoms) up to a $8 \times 8 \times 8$ (6144-atoms). The electronic structure of fully relaxed structures (vc-relax) was also computed, including spin-orbit coupling (SOC) and PBE0. The supercell distortion with respect to the perfect cubic α-FAPbI3 has been estimated by the mean squared error (MSE) between the lattice vectors of the two systems. Because CP2K[59] allows to run also DFT at 0 K, Γ-point simulations were performed with both the QE and CP2K software, obtaining equal band gap values to three decimal places, which means that the results achieved with the two software packages are comparable.

## Ab initio molecular dynamics

AIMD simulations were run in the DFT framework as implemented in the CP2K software. The PBE and PBE0 functional and the D3 dispersion correction[60] were adopted together with Goedecker-Teter-Hutter pseudopotentials[61] and a polarized double-ζ Gaussian basis set (DZVP)[62] for valence electrons. The energy cutoff for the expansion of the electron density was set to 400 Ry. Simulations were run with a time step of 0.5 fs in the NPT flexible ensemble using Born-Oppenheimer dynamics for 7–12 ps (PBE) and 2–5 ps (PBE0), while the temperature was controlled by the Bussi thermostat[63] and the pressure by the Martyna barostat[64]. Different simulation cells were used, starting from a $1 \times 1 \times 1$ α-phase FAPbI3 (12-atoms) up to a $8 \times 8 \times 8$ (6144-atoms). The finite temperature band gap was computed as an average of different band gaps calculated from the projected density of states (PDOS) on several AIMD snapshots after the system equilibrated (∼2 ps). The spatial variation of the band gap within a supercell was calculated by grouping the PDOS of the atoms of interest. Real space positions of the top of the valence and bottom of the conduction bands in FAPbI3 were computed after quenching an equilibrated AIMD snapshot to 0 K. Different initial FA orientations were tested - completely ordered, randomly oriented, and smart oriented (total FA dipole equal to zero) - to avoid any bias in the simulations. The total dipole moment was calculated at the quantum level as in the CP2K framework, while the contribution of FAs to the dipole in the initial configurations was estimated classically, assigning a + 1 charge to each FA.

## Pseudo-randomly oriented FA configurations and classical dipole moment

In order to generate a starting configuration with pseudo-randomly oriented FAs, an in-house Python code has been developed. This code, when provided with a perfect cubic perovskite cell of a specified size, generates different configurations with pseudo-randomly oriented FA molecules. Among these, the one with the lowest overall dipole moment is selected. This dipole moment is calculated classically by assigning charges +2 and −1 to Pb and I, respectively. For the FA, the dipole vector is oriented along the C-H axis, and the partial charges of the atoms are taken from our in-house fixed point-charge force field for FAPbI3 in a way that the total sum equals +1 (Table 4). This force field was trained on AIMD trajectories and stabilizes both α- and δ-FAPbI3 at the experimentally observed temperature ranges. The pseudo-random orientations are chosen in such a way that the C-H axis of the FAs aligns with one of the four 3-fold symmetry axes of the cubic $PbI_6$ cage, namely [111], [$\bar{1}$11], [1$\bar{1}$1], and [11$\bar{1}$] while N-N axis is rotated in such a way as to enable hydrogen bonding between one of the N-H to one of the I$^-$ of the $PbI_6$. The specific choice of one of the four axes for the C-H orientation and the specific I to which one of the N-H hydrogen bonds is determined by random generation.

## Time correlation function analysis

The rotational dynamics of FA and $PbI_6$ octahedra were characterised by the correlation function

$$C_{rot}(t) = \frac{\langle \boldsymbol{\mu}(t) \cdot \boldsymbol{\mu}(0) \rangle}{|\boldsymbol{\mu}(0)|} \quad (1)$$

Where $\boldsymbol{\mu}(t)$ is the C-H(N-N) vector for FA or the octahedra tilting function appropriate for the $PbI_6$ tilting. The timescale oscillations for the band gap were quantified by the correlation function

$$C_{gap}(t) = \frac{\langle \Delta\epsilon_{cv}(t) \cdot \Delta\epsilon_{cv}(0) \rangle}{|\Delta\epsilon_{cv}(0)|} \quad (2)$$

where $\Delta\varepsilon_{cv}(t)$ is the difference between the eigenvalues of the bottom of the conduction band and the top of the valence band. The same time correlation function has been used to compute the correlations of the dipole moment fluctuations, where the quantity correlated in time was the value of the dipole moment.

## Data availability

All the data generated in this study have been deposited in the Zenodo database at https://doi.org/10.5281/zenodo.13712682. All the data used in this study to generate the figures and tables are provided in the Supplementary Information/Source Data files.

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

## Acknowledgements

U.R. acknowledges the Swiss National Foundation (grant No. 200020_219440) and computational resources from the Swiss National Computing Centre CSCS. V.C. acknowledges computational resources from the Swiss National Computing Centre, CSCS.

## Author contributions

V.C. and L.A. conceived the idea. V.C., L.A., V.S., A.V., and N.L. performed the theoretical simulations and wrote the paper under the supervision of U.R. All authors contributed to the discussion and writing of the paper.

## Competing interests

The authors declare no competing interests.
