## [Transparent Peer Review file · Nature Communications]

Nanoscale size effects in α -FAPbI₃ evinced by large-scale ab initio simulations

Corresponding Author: Professor Ursula Rothlisberger

Version 0:

Reviewer comments:

Reviewer #1

(Remarks to the Author)

This paper investigates the effect of the size of the simulated system on the accurate description of the structural and electronic properties in FAPbI₃ by using large-scale first-principles calculations. This is an important material for the community working on halide perovskite solar cell materials and these results are of interest. The paper could increase its broad appeal by discussing more the "why" and comparing FAPbI₃ to related perovskite systems, such as MAPbI₃, which has more polar MA cations, and CsPbI₃, which contains the non-polar cations. Would the observed size effect vary for these systems? particularly in terms of dipole moment properties. If no significant difference is expected, what factors might explain the uniformity across these materials?

There are also technical comments that need to be addressed before publication:

(1) It's a bit surprise and the correct bandgap (1.64-1.68 eV) of FAPbI₃ can be reproduced using HSE-SOC with a default mixing rate of 0.25. Various previous studies showed that the bandgap remains underestimated with a 0.25 mixing, and typically a mixing rate ~0.55 is necessary, yielding a band gap of only ~1.50 eV (e.g., Phys. Rev. Lett. 128, 136401 (2022)). This difference should be addressed. Are structures relaxed with spin orbit? Moreover, it shows a considerably large bandgap for the unicell model whatever the functional scheme, even with 2x2x2 k-points. I would suggest re-checking the perovskite structure used in the calculations, as this could influence the reliability of the results.

(2) The claim on page 12 that "from a 2592-atoms cell upwards, the PBE approximation is already able to describe the electronic band gap of α -FAPbI₃". While the band gap value predicted by the pure PBE scheme for hybrid perovskites often aligns well with experimental data, the absolute positions of the valence band maximum (VBM) and conduction band minimum (CBM) obtained using this method can deviate significantly from their actual positions due to large self-interaction error (e.g., J. Phys. Chem. Lett. 2015, 6, 1461–1466). Such an artifact may lead to qualitative inaccuracies in certain fundamental properties, such as defect (deep)-level prediction. This issue should be appropriately addressed. To ensure a more accurate theoretical description, the positions of the valence and conduction band edges should be used as an additional criterion alongside bandgap values, rather than relying on the bandgap value alone.

(3) The methods section should provide more details on the procedure used to generate the randomly oriented FAPbI₃ structure. Was this done manually, or was a specific algorithm or software used? One has to be careful in these systems to break symmetry to find true ground state configurations.

(4) There is a typo in Table 2—'3000K' should be corrected.

Reviewer #2

(Remarks to the Author)

The paper mainly discusses the importance of cell size in obtaining correct bandgap values, and zeroing out the total dipole moment in FAPbI₃ with different Density Functional Theory-functional. It is recommended to explain the result in a more detailed way. In some cases, it is assumed that something is very obvious to the readers, which is not the case. The results give a view about using the right theoretical description while comparing with experimental results, however more detailed explanation with more supporting information is necessary to clarify. The authors used standard Density Functional Theory to calculate the electronic structure calculations in the given material. The provided calculation details are enough to reproduce the results, however, it is recommended to give the structural details of the considered phase in supporting information. There are a few comments in the following, which may improve the paper.

Comments:

1. "Several computational methods have been adopted to determine the band gap of FAPbI₃, spanning from density functional theory (DFT) with different generalized gradient approximations, meta functionals or (range-separated) (meta) hybrid functionals, van der Waals functionals, to the GW approximation of many-body perturbation theory, or combinations of them."----- Please include some references to validate the claim.
2. "In addition, spin-orbit coupling (SOC) is usually implemented in presence of Pb/Sn atoms, which leads to a significant lowering of the band gap."---Please include some references.
3. The authors considered two types of configurations: all-aligned and pseudo-random. When vc-relax is done, how did they constrain the alignment of FAs in the first case?
4. "For both the relax and vc-relax calculations, the band gap converges with increasing k-point grid but to different final values, indicating that the relaxation of all the cell degrees of freedom is necessary"---Please elaborate on what you mean by relaxation of all the cell degrees of freedom.
5. What is the reason behind having a large MSE for the 96-atom cell in Table 2?
6. "In contrast, the structures optimised with all-aligned FAs do not present an octahedral tilting pattern."---What is the value that corresponds to tilt angles with the all-aligned configuration in Table 2, then?
7. "A comparison of all aligned and pseudo-random configurations reveals that the vc-relax band gap is only improved for the 2592-atoms (3×3×3) cell"---This statement is not clear. Please elaborate.
8. The sequence in explaining the result is very confusing. The authors started to explain the result of Table 1, then jumped to Table 2, and again came back to Table 1. The sequence of presenting the tables needs to be improved for the sake of the reader.
9. "It has been observed that, for all supercells, the potential energy with pseudo-randomly oriented FAs is consistently lower than that of the all-aligned case."---Please add a figure in the supporting information justifying the statement.
10. "By adding SOC, the band gaps of the 12- and 96-atoms cells converge to slightly different values - 0.45 153 eV and 0.54 eV....."---is it a negative sign before 0.45, or a typo?
11. In Table 2, the second value corresponds to octahedra tilt angles for 6188 atoms referring to pseudo-randomly oriented FA? If yes, the * is missing there.
12. Also in Table 2, the heading should be "Total Dipole PBE0 300 K (Debye/ABX₃)", and not "Total Dipole PBE0 3000 K (Debye/ABX₃)".
13. "This again indicates the importance of the system size, in order to detect the possible formation of band gap domains."---Is it possible to show some convergence with size of the cell in supporting information? Did the author try going beyond the 6144-atom cell? Would that give the same results?
14. Values reported in Table 3 are very confusing. It is not well explained that how the dipoles are calculated. In the third column, it says "Total dipole-FA dipole", which does not match the values given in the column. Please explain how do you get the values.

Reviewer #3

(Remarks to the Author)

This manuscript presents a comprehensive computational investigation of size effects in α -FAPbI₃ perovskite using large-scale ab initio simulations. The authors employ both static DFT calculations and ab initio molecular dynamics to systematically examine how system size affects structural and electronic properties, with particular focus on band gap, structural distortions, and dipole moments. The authors present a systematic study employing multiple computational approaches and progressively larger system sizes, from 12 to 6144 atoms. Their analysis effectively establishes connections between structural features, electronic properties, and dipole moments.

However, several significant concerns need to be addressed. While the work is technically sound, similar conclusions about system size effects have been previously demonstrated for other hybrid perovskites, particularly MAPbI₃. The authors should more clearly articulate what new physical insights their study provides beyond confirming similar behavior in FAPbI₃. More importantly, the PBE0+SOC calculations, which shown to be the most accurate approach, are only performed for systems up to 96 atoms. While the computational cost of such calculations is acknowledged, this limitation undermines the main conclusion that "the size of the simulated system needs to approach the nanoscale." The authors should either extend these calculations to larger systems, provide evidence of convergence at 96 atoms, or explicitly discuss this limitation and its implications for their conclusions.

The manuscript could be suitable for publication after addressing these concerns, particularly by adding explicit

comparisons with previous MAPbI₃ studies and providing a more thorough analysis of PBE0+SOC convergence. The authors should also more explicitly discuss the limitations of using PBE results for larger systems.

Version 1:

Reviewer comments:

Reviewer #1

(Remarks to the Author)

The authors have revised the manuscript with new discussion and additional calculations. I have minor reservations, I wish the revised manuscript elaborated on how and why the size effect varies across these prototypical perovskites. A more explicit and in-depth analysis would enhance the novelty and rigor of the study beyond the standard expectations. After that, I have no further comments and the manuscript can be considered acceptable for publication.

Reviewer #2

(Remarks to the Author)

I have reviewed the revised manuscript and the authors' responses to my comments. I appreciate the effort the authors have put into addressing the points raised in the initial review. They have provided clear and satisfactory responses to all my queries, and the revisions improved the manuscript.

Reviewer #3

(Remarks to the Author)

The authors adequately responded to the comments and revised the article accordingly.

Version 2:

Reviewer comments:

Reviewer #1

(Remarks to the Author)

The authors have addressed all the issues. The current form will be a good presentation, and I recommend acceptance in Nature Communications without further revision.

Response to referee comments

Virginia Carnevali,¹ Lorenzo Agosta,¹ Vladislav Slama,¹
Nikolaos Lempesis,¹ Andrea Vezzosi,¹ and Ursula Rothlisberger¹

¹*Laboratory of Computational Chemistry and Biochemistry,
Institute of Chemical Sciences and Engineering,
Swiss Federal Institute of Technology (EPFL), Lausanne, Switzerland*

Dear Editor,

Please find enclosed a revised version of our manuscript with the title “Nanoscale size effects in α -FAPbI₃ evinced by large-scale ab initio simulations”. The authors would like to thank the referees for their pertinent comments and suggestions, which significantly helped to improve the quality of this manuscript. We have revised the manuscript carefully and thoroughly. Changes in the manuscript itself are labeled in red. In the following, we present detailed itemized responses/corrections to all the referee comments, first summarizing the main changes made in the revised manuscript. Here, all author responses are colored in blue while additions to the manuscript and Supporting Material are in red.

Response to Referee #1

This paper investigates the effect of the size of the simulated system on the accurate description of the structural and electronic properties in FAPbI₃ by using large-scale first-principles calculations. This is an important material for the community working on halide perovskite solar cell materials and these results are of interest.

We thank the referee for recognizing and appreciating the importance of our work.

The paper could increase its broad appeal by discussing more the “why” and comparing FAPbI₃ to related perovskite systems, such as MAPbI₃, which has more polar MA cations, and CsPbI₃, which contains the non-polar cations. Would the observed size effect vary for these systems? particularly in terms of dipole moment properties. If no significant difference is expected, what factors might explain the uniformity across these materials?

We agree with the referee and thank them for this suggestion. We have expanded the introduction in order to further clarify the main aim of this work and the differences

with MAPbI₃ and CsPbI₃. Studies on MAPbI₃ (DOI: [dx.doi.org/10.1021/nl503494y](https://doi.org/10.1021/nl503494y) and [10.1016/j.joule.2023.03.017](https://doi.org/10.1016/j.joule.2023.03.017)) and CsPbI₃ (DOI: <https://doi.org/10.1002/sml.202303565>) have also pointed out that there is a dependence of the electronic properties on system size, mainly induced by the phase transition from orthorhombic to tetragonal and cubic (reflected in a change of tilting angle between PbI₆ octahedra). How and why these local distortions affect the macroscopic band gap of the material and what is the relation between system size and electronic properties are partially discussed in the literature. This is mainly due to the computational limits present in ab initio simulations of cells containing more than 500-800 atoms. Furthermore, some confusion was generated by trying to explain the experimental band gap observed for MAPbI₃, CsPbI₃, and FAPbI₃ using small unit cells albeit at high levels of theory. We show here that the system size is actually the key parameter for an accurate description of the electronic properties. Moreover we illustrate by direct calculations of the band gaps of systems with large systems sizes that the macroscopically measured band gap is the result of an average of local band gaps induced by the fluctuations in octahedral tilting. The length and time scales of this correlations are driven by the FA cation dynamics. We have added a couple of references in the introduction where we already discussed about MAPbI₃ and CsPbI₃. A review of the extant literature indicates that, for MAPbI₃, a behavior analogous to that exhibited by FAPbI₃ is to be anticipated, given a similar system size. CsPbI₃ also requires a large system size to recover the octahedra buckling properly, while the three-fold symmetry of the FA and MA cations is lowered to the two equivalent positions that Cs can take inside the PbI₆ cages. This releases the constraints on the number of cell repetitions to only even numbers.

Regarding “why our study is important”, we have included these considerations in the “Introduction” section of the revised manuscript.

“Despite experimental and computational insights that point to a dependence of the electronic band gaps of MHPs and the system size, a direct assessment of this effect is missing. Computer simulations in the context of ab initio molecular dynamics would be able to tackle the complexity of this challenge and help to explain the experiments by linking electronic structure and size effects. However, the limitations imposed in length and time scales in by ab initio calculations discouraged the exploration of large systems in the field of MHPs, favoring simulations with small supercells with high level of theory as previously mentioned.”

1. It's a bit surprise and the correct bandgap (1.64-1.68 eV) of FAPbI₃ can be reproduced using HSE-SOC with a default mixing rate of 0.25. Various previous studies showed that the bandgap remains underestimated with a 0.25 mixing, and typically a mixing rate 0.55 is necessary, yielding a band gap of only ~ 1.50 eV (e.g., Phys. Rev. Lett. 128, 136401 (2022)). This difference should be addressed. Are structures relaxed with spin orbit? Moreover, it shows a considerably large bandgap for the unicell model whatever the functional scheme, even with 2x2x2 k-points. I would suggest re-checking the perovskite structure used in the calculations, as this could influence the reliability of the results.

We are not sure to what HSE-SOC results the reviewer is referring to since we did not present any HSE or HSE+SOC calculations in this paper and literature values for HSE(-SOC) vary enormously (e.g. <https://doi.org/10.1039/d0ra06028c>). In the reference the reviewer mentions (Phys. Rev. Lett. 128, 136401 (2022)), a band gap of 1.50 eV is indeed computed with HSE+SOC with a mixing parameter of 0.55 for a $\sqrt{2} \times \sqrt{2} \times 1$ tetragonal unit cell of FAPbI₃ (96 atoms) without specifying the k-point grid. In the Supporting Information, the authors refer to another paper (X. Zhang, J.-X. Shen, M. E. Turiansky, and C. G. Van de Walle, Nat. Mater. 20, 971 (2021)) for the computational details but in that paper the band gap is computed for a tetragonal unit cell of FAPbI₃ with a k-point grid of $3 \times 3 \times 1$. But, as we show in Table 1 of the manuscript, even with a 96-atom cell, increasing the k-point grid is important for converging the band gap. Before tuning the HSE mixing parameter, one should ensure convergence otherwise the results may not be reliable.

We could not run PBE0 and PBE0+SOC for the largest cells because the calculations required too much memory. However, if we assume an effective PBE0-SOC cancellation, the 2592-atoms cell with a band gap of 1.50 eV is our most accurate estimate of the correct theoretical band gap.

In order to clarify this point, we have added the following paragraph to the main manuscript:

“It was not feasible to run PBE0+SOC for systems larger than 96-atoms due to the excessive memory requirements of the calculations. However, making use of the observed cancellation between PBE0 and SOC effects, the 2592-atoms cell with a PBE band gap of 1.50 eV (Tab.1) recovers the experimental band gap.”

In order to avoid confusion, we have also reformulated the sentence “It is known that for elements such as Pb, SOC and PBE0 contributions should cancel each other out.” to “It

is known that for materials such as lead halide perovskites, SOC and PBE0 contributions should cancel each other out.”.

2. The claim on page 12 that “from a 2592-atoms cell upwards, the PBE approximation is already able to describe the electronic band gap of α -FAPbI₃”. While the band gap value predicted by the pure PBE scheme for hybrid perovskites often aligns well with experimental data, the absolute positions of the valence band maximum (VBM) and conduction band minimum (CBM) obtained using this method can deviate significantly from their actual positions due to large self-interaction error (e.g., J. Phys. Chem. Lett. 2015, 6, 1461-1466). Such an artifact may lead to qualitative inaccuracies in certain fundamental properties, such as defect (deep)-level prediction. This issue should be appropriately addressed. To ensure a more accurate theoretical description, the positions of the valence and conduction band edges should be used as an additional criterion alongside bandgap values, rather than relying on the bandgap value alone.

The referee is correct and we are grateful for this suggestion. The positions of the valence band maximum (VBM) and conduction band minimum (CBM) obtained using PBE can differ significantly from the ones obtained at the PBE0+SOC level of theory. In addition, the symmetry of the VBM and CBM states are not identical because SOC induces a band splitting. We have now included a new table in the Supporting Information (Table S1) that shows the absolute energies of the states at VBM and CBM for each band gap value computed with the (0 K) first-principles calculations reported in Table 1 of the manuscript. The eigenvalues of the VBM and CBM generally parallel the trends observed for the band gap convergence. The absolute values of the VBM and CBM are affected by SOC and PBE0 (Tab.S1) in agreement with the picture reported in J. Phys. Chem. Lett. 2015, 6, 1461-1466, even when PBE0 and SOC cancel each other out for their difference (96-atom cell with k-point grid converged).

To make this point clear, we have added the following paragraph at page 6 of the main manuscript:

“Even though PBE0 and SOC effects cancel each other out once the system size reaches convergence, thereby leaving the band gap unchanged with respect to the PBE *vc-relax* value, PBE0+SOC values offer a more accurate estimation of the absolute position of the valence band maximum (VBM) and conduction band minimum (CBM). This enhanced accuracy

can be attributed to a reduction in self-interaction error and an improved description of the asymptotic behavior of the exchange-correlation potential, which is of paramount importance for the calculation of correct band alignments and the estimation of defect formation energies⁴⁹. The symmetry of the band edge states also changes because of the band splitting induced by SOC. The (0 K) Kohn-Sham eigenvalues of the VBM and CBM obtained for fully optimized structures are reported in Tab.S1. At the PBE0+SOC level of theory, we obtain a lowering of 0.6 eV of the PBE Kohn-Sham eigenvalues even if the PBE0 and SOC effects cancel each other out for the band gap.”

and at page 13 of the main manuscript:

“The evaluation of the absolute positions and symmetry of VBM and CBM with PBE will be affected by a significant self-interaction error and the wrong asymptotic behavior of the exchange-correlation potential. These errors can be reduced by performing a single-point energy calculation on the PBE optimized structures using PBE0+SOC. Despite the fact that PBE0 and SOC effects cancel each other out for the band gap as often observed for lead halide perovskites, these corrections provide a more accurate estimation of the absolute values of the Kohn-Sham eigenvalues. ”

We have also added a new reference:

- Du, M.-H. Density Functional Calculations of Native Defects in $\text{CH}_3\text{NH}_3\text{PbI}_3$: Effects of Spin-Orbit Coupling and Self-Interaction Error. *J. Phys. Chem. Lett.* 6(8), 1461-1466 DOI: 10.1021/acs.jpcllett.5b00199 (2015).

A table with the absolute energy eigenvalues of VBM and CBM has been added to the SI:

Table S1. FAPbI₃ Kohn-Sham valence band maximum (VBM) and conduction band minimum (CBM) for different system sizes at several theory levels and system optimization schemes. The initial configurations were chosen with all FAs aligned except the ones indicated with a star superscript.

Simulation cell	relax PBE 0 K (eV)		vc-relax PBE 0 K (eV)		vc-relax PBE+SOC 0 K (eV)		vc-relax PBE0 0 K (eV)		vc-relax PBE0+SOC 0 K (eV)		$n \times n \times n$ k-point grid
	VBM	CBM	VBM	CBM	VBM	CBM	VBM	CBM	VBM	CBM	
12-atom (1 × 1 × 1)	1.70	5.38	2.61	6.33	2.86	6.32	1.53	7.35	1.80	7.34	1
	2.31	4.51	1.90	4.20	2.02	3.79	0.90	4.81	1.04	4.37	2
	2.53	4.02	1.81	3.78	1.94	3.18	0.89	4.35	1.00	3.72	4
	2.53	4.03	2.30	3.93	2.50	2.99	1.18	4.62	1.40	3.68	6
	2.53	4.03	2.31	3.89	2.51	2.99	1.19	4.57	1.41	3.66	8
	2.53	4.02	2.31	3.86	2.51	2.96	1.19	4.56	1.41	3.66	10
96-atom (2 × 2 × 2)	2.30	4.54	2.31	4.88	2.09	3.88	1.25	4.61	1.38	4.24	1
	2.49	4.04	2.22	3.91	2.41	3.02	1.42	4.27	1.63	3.27	2
	2.53	4.02	2.30	3.89	2.51	3.00	1.51	4.24	-	-	4
	2.54	4.02	2.28	3.89	2.48	3.02	1.67	4.42	-	-	6
	2.45*	4.06*	2.46*	4.06*	2.65*	3.13*	_*	_*	_*	_*	6
768-atom (4 × 4 × 4)	2.50	4.04	2.22	3.83	2.44	2.92	-	-	-	-	1
	2.38*	4.12*	2.38*	4.12*	2.57*	3.20*	-	-	-	-	1
	2.31	3.88	2.31	3.88	2.51	2.96	-	-	-	-	2
	2.42*	4.11*	2.42*	4.11*	2.61*	3.17*	-	-	-	-	2

* FA pseudo-randomly oriented

3. The methods section should provide more details on the procedure used to generate the randomly oriented FAPbI₃ structure. Was this done manually, or was a specific algorithm

or software used? One has to be careful in these systems to break symmetry to find true ground state configurations.

As mentioned in the submitted manuscript and shown in Fig. S1, in the pseudo-random systems, the FAs are oriented along the 3-fold symmetry axes of the PbI_6 octahedra, i.e. the C-H axis of the FAs is always aligned with the four 3-fold symmetry axes $[111]$, $[\bar{1}11]$, $[1\bar{1}1]$, and $[11\bar{1}]$; while the N-N axis is aligned in such a way as to enable hydrogen bonding of one of the N-H to one of the I^- of the PbI_6 . The random part of these FA placements is which of the 4 axes is chosen and to which I^- one of the N-H points to. To generate these configurations, we used an inhouse Python code that, given a perfect cubic perovskite cell of a given size, generates a set of pseudo-randomly oriented FA configurations, among which the one with the lowest dipole moment is selected. The overall dipole moment is calculated classically by assigning charges +2 and -1 to Pb and I, respectively. For the FA, the dipole vector is pointing along the C-H axis (directed from H to C in positive-to-negative convention) and the partial charges of the atoms are taken from our inhouse fixed point-charge force field for FAPbI_3 with the total sum equal to +1. This force field was trained on AIMD trajectories and stabilizes both α - and δ - FAPbI_3 at the respective temperatures. The used charges are now listed in a new table in the “Methods” section (Table 4).

After relaxation of both atomistic and cell degrees of freedom, an energetic comparison between the all-aligned and pseudo-randomly oriented FA configurations indicates that the pseudo-random one is always the more stable configuration.

To clarify how the pseudo-randomly oriented FA configurations are built, we have added the following paragraph in the “Methods” section of the main manuscript:

“ Pseudo-randomly oriented FA configurations and classical dipole moment

In order to generate a starting configuration with pseudo-randomly oriented FAs, an inhouse Python code has been developed. This code, when provided with a perfect cubic perovskite cell of a specified size, generates different configurations with pseudo-randomly oriented FA molecules. Among these, the one with the lowest overall dipole moment is selected. This dipole moment is calculated classically by assigning charges +2 and -1 to Pb and I, respectively. For the FA, the dipole vector is oriented along the C-H axis, and the partial charges of the atoms are taken from our inhouse fixed point-charge force field for FAPbI_3 in a way that the total sum equals +1 (Table 4). This force field was trained on AIMD trajectories

and stabilizes both α - and δ -FAPbI₃ at the experimentally observed temperature ranges. The pseudo-random orientations are chosen in such a way that the C-H axis of the FAs aligns with one of the four 3-fold symmetry axes of the cubic PbI₆ cage, namely [111], [$\bar{1}\bar{1}\bar{1}$], [$1\bar{1}\bar{1}$], and [$11\bar{1}$] while N-N axis is rotated in such a way as to enable hydrogen bonding between one of the N-H to one of the I⁻ of the PbI₆. The specific choice of one of the four axes for the C-H orientation and the specific I to which one of the N-H hydrogen-bonds are determined by random generation.”

Atom	Charge	Description
C	-0.06	-
N	-0.19	-
H _C	+0.27	H bonded to C
H _{NC}	+0.28	H bonded to N closer to C
H _{NN}	+0.30	H bonded to N further from C
Pb	+2	-
I	-1	-

Table 4. Partial charges for Pb, I, C, N, and H atoms taken from our in-house point charge force field for FAPbI₃. These charges were used for computing the classical dipole moment of the supercells.

We have also revised Fig.S1.

4. There is a typo in Table 2—'3000K' should be corrected.

We thank the referee for catching this typo that we have now corrected.

Response to Referee #2

The paper mainly discusses the importance of cell size in obtaining correct bandgap values, and zeroing out the total dipole moment in FAPbI₃ with different Density Functional Theory-functional. It is recommended to explain the result in a more detailed way. In some cases, it is assumed that something is very obvious to the readers, which is not the case. The results give a view about using the right theoretical description while comparing with experimental results, however more detailed explanation with more supporting information

Figure S1. (a) The four 3-fold axes of the cubic symmetry. (b) Orientations of the FA molecule such that its molecular dipole satisfies the 3-fold symmetry within the PbI_6 cage. I, Pb, C, N, and H are shown in violet, white octahedra, brown, light blue, and pink, respectively.

is necessary to clarify. The authors used standard Density Functional Theory to calculate the electronic structure calculations in the given material. The provided calculation details are enough to reproduce the results, however, it is recommended to give the structural details of the considered phase in supporting information. There are a few comments in the following, which may improve the paper.

We thank the referee for appreciating our work while suggesting improvements in the presentation of the results. We have carefully considered each comment and provided further explanation and/or analysis as highlighted in the point-by-point responses to the referee's comments below.

1. "Several computational methods have been adopted to determine the band gap of FAPbI_3 , spanning from density functional theory (DFT) with different generalized gradient approximations, meta functionals or (range-separated) (meta) hybrid functionals, van der Waals functionals, to the GW approximation of many-body perturbation theory, or combinations of them." — Please include some references to validate the claim.

We have added the following references to the revised version of the manuscript:

- Zhao, X.G. et al. Polymorphous nature of cubic halide perovskites. *Phys Rev B*. 101: 1-19 DOI: <https://doi.org/10.1103/PhysRevB.101.155137> (2020).
- Hernández-Haro, N. et al. DFT prediction of band gap in organic-inorganic metal halide perovskites: an exchange-correlation functional benchmark study. *Chem Phys*. 516: 225-231 DOI: <https://doi.org/10.1016/j.chemphys.2018.09.023> (2019).
- Muhammad, Z. et al. I. Tunable relativistic quasiparticle electronic and excitonic behavior of the FAPb(I_{1-x}Br_x)₃ alloy. *Phys Chem Chem Phys*. 22: 11943-11955 DOI: <https://doi.org/10.1039/D0CP00496K> (2020).
- Pan, Y.Y. et al. First-Principles Study on Electronic Structures of FAPbX₃ (X = Cl, Br, I) Hybrid Perovskites. *J Adv Nanomater.* 1: 1 DOI: <https://dx.doi.org/10.22606/jan.2016.11004> (2016).
- Filip, M.R. et al. Steric engineering of metal-halide perovskites with tunable optical band gaps. *Nat Commun.* 5: 1-9 DOI: <https://doi.org/10.1038/ncomms6757> (2014).
- Li D, Meng J, Niu Y, Zhao H, Liang C. Understanding the low-loss mechanism of general organic–inorganic perovskites from first-principles calculation. *Chem Phys Lett*. 627: 13-19 DOI: <https://doi.org/10.1016/j.cplett.2015.03.028> (2015).
- Zeeshan, M. et al. Revealing the quasiparticle electronic and excitonic nature in cubic, tetragonal, and hexagonal phases of FAPbI₃. *AIP Advances* 12 (2): 025330 DOI: <https://doi.org/10.1063/5.0076738> (2022).

2. “In addition, spin-orbit coupling (SOC) is usually implemented in presence of Pb/Sn atoms, which leads to a significant lowering of the band gap.”—Please include some references.

In addition to the previous references, we have added:

- Valadares, F. et al. Electronic structure panorama of halide perovskites: approximated DFT-1/2 quasiparticle and relativistic corrections. *J Phys Chem C*. 124: 18390-18400 (2020).

3. The authors considered two types of configurations: all-aligned and pseudo-random. When vc-relax is done, how did they constrain the alignment of FAs in the first case?

The all-aligned FA configurations are built by replicating a perfect $1 \times 1 \times 1$ cubic α -FAPbI₃ cell. The resulting supercell has a perfect cubic lattice of PbI₆ octahedra and FAs all parallel to each other. This is a highly symmetric metastable state that even vc-relax cannot break. After vc-relax, the FAs are still all aligned, while the cubic lattice of PbI₆ octahedra has been squeezed along one of the $[111], [\bar{1}11], [1\bar{1}1], [11\bar{1}]$ directions to compensate for the dipole moment created by the all-aligned FAs.

4. “For both the relax and vc-relax calculations, the band gap converges with increasing k-point grid but to different final values, indicating that the relaxation of all the cell degrees of freedom is necessary”—Please elaborate on what you mean by relaxation of all the cell degrees of freedom.

We mean that both the atomic coordinates and the cell degrees of freedom (lattice constants and angles) must be relaxed, which means that the relax calculation (atomic coordinates only) is not enough, but vc-relax is needed. Indeed, if the relax calculation would be sufficient, the band gap computed with relax should be the same as the one computed with vc-relax.

We have now explained this in the revised version of the manuscript by replacing “For both the relax and vc-relax calculations, the band gap converges with increasing k-point grid but to different final values, indicating that the relaxation of all the cell degrees of freedom is necessary” with “For both the relax and vc-relax calculations, the band gap converges with increasing k-point grid but to different final values, **indicating that both atomic positions and lattice parameters, i.e. lattice constants and angles (vc-relax) must be relaxed.**”

5. What is the reason behind having a large MSE for the 96-atom cell in Table 2?

The 96-atom cell (made up by $2 \times 2 \times 2$ unit cells) is the one that differs the most from the perfect cubic structure after vc-relax. This can be explained by the fact that the number of unit cells does not correspond to a multiple of 3, which makes this system the least compatible among the different investigated supercells for accommodating the 3-fold orientational symmetry of the FA molecules. As a consequence, none of the possible pseudo-randomly oriented FA configurations of the 96-atom cell can completely cancel out the dipole moment of the system. One could object that the 12-atom ($1 \times 1 \times 1$) cell should be even worse for the dipole moment, and this is indeed the case from our Tab.1 of the main manuscript, however this does not result in an even more pronounced distortion with respect to the cubic

structure since the system is very small and consequently there are less possibilities for the atoms to move, i.e. the system is more rigidly constraint.

6. “In contrast, the structures optimised with all-aligned FAs do not present an octahedral tilting pattern.”—What is the value that corresponds to tilt angles with the all-aligned configuration in Table 2, then?

For the all-aligned FA configurations, after vc-relaxation, the octahedra do not show the usual buckling pattern along the x,y,z axes, but they are all stretched along one of the $[111], [\bar{1}\bar{1}\bar{1}], [1\bar{1}\bar{1}], [11\bar{1}]$ directions to compensate for the dipole moment created by the all-aligned FAs, as shown in Fig. S3 of the original manuscript. This means that in the case of all-aligned FAs, the tilt angle indicates the maximum rotation of the octahedra with respect to one of the Cartesian axes. In contrast to the pseudo-randomly oriented FAs, where the octahedral rotation alternates between θ and $-\theta$ tilting angles, for the all-aligned FAs all tilting angles have the same sense of rotation in θ .

We have now improved the caption of Tab.2:

“Table 2. Mean squared error (MSE) of the lattice vectors with respect to the perfect cubic FAPbI_3 α -phase (column 1) and absolute maximum value of the octahedral tilting angle (column 2) at 0 K. For the structure optimized in the presence of all-aligned FAs, the octahedral tilting angle refers to the global deformation of the octahedra, **resulting in a tilting angle of θ for all octahedra**; while for the structure optimized from pseudo-randomly oriented FAs **the tilting angle** refers to the average absolute value of **the typical $-\theta/\theta$ tilting pattern** (Supplementary Fig.3). Total dipole moment ($|\mathbf{d}_{Tot}|$) per stoichiometric unit (ABX_3) for different simulation cells; the dipole moment was computed at 0 K (column 3) and after equilibration as average along the AIMD trajectories performed at 300 K in the NPT-F ensemble (columns 4-5). The initial configurations for the 0 K simulations were chosen with all FAs aligned except the ones indicated with a star superscript.”

and Fig.S3:

7. “A comparison of all aligned and pseudo-random configurations reveals that the vc-relax band gap is only improved for the 2592-atoms ($3\times 3\times 3$) cell”—This statement is not clear. Please elaborate.

Only for the 2592-atom cell, when compared with the other supercells, the vc-relax band gap of the pseudo-random configuration is lower (and compatible with the experimental one)

Figure S3. Zoom on the final FAPbI₃ structures starting from all-aligned (a) and pseudo-random (b) FA configurations after vc-relax. The Pb-I distances for two adjacent octahedra are highlighted. The typical $-\theta/\theta$ tilting pattern of the octahedra can be seen in the pseudo-random case, while the all-aligned case shows a non-physical collective I-shift resulting in a tilting angle of θ for all octahedra. The reference system for the tilt angle and the tilting angles are shown in red. The color code is the same as in Fig.1.

than that of the all-aligned one. The reason for this, as explained in the main text, is that this supercell accommodates both the 3-fold symmetry of the FAs and the symmetry of the octahedral tilting pattern, which requires an even number of unit cells. We noticed that we wrongly referred to the 2592-atom cell as $3 \times 3 \times 3$, while the correct number is of course $6 \times 6 \times 6$. We apologize for this and corrected it in the revised version of the manuscript.

8. The sequence in explaining the result is very confusing. The authors started to explain the result of Table 1, then jumped to Table 2, and again came back to Table 1. The sequence of presenting the tables needs to be improved for the sake of the reader.

We understand that it might not be easy to jump from one table to the other and then back to the first one, but we think that it is better to keep one table for the calculated band gaps both at 0 K and at 300 K, and one for the structural properties and the dipole moment again at 0 K and at 300 K. This table format allows to compare the single properties without and with temperature effects. Combining all the information of the two tables into a single one would be too much. Regarding the presentation of the results, we believe that keeping the characterization of α -FAPbI₃ at 0 K and 300 K separate is better for a more fluent narrative.

We have thought carefully about how to present the results, and we hope that the additions made in the revised version in response to the referees' requests/comments will improve the readability of the manuscript.

9. “It has been observed that, for all supercells, the potential energy with pseudo-randomly oriented FAs is consistently lower than that of the all-aligned case.”—Please add a figure in the supporting information justifying the statement.

As suggested by the referee, we have added Figure S4 in the Supplementary material to support our statement.

Figure S4. Comparison of the total energy per unit cell between the all-aligned and pseudo-randomly oriented FA configurations for different supercells. On the x-axis, the number of atoms in the supercell and the k-point grid (e.g., k1=1 × 1 × 1 k-point grid) are given. The all-aligned FA configurations (gray) are always higher in energy (less stable) and are set to 0 eV for each supercell. The energies of the pseudo-randomly oriented FA configurations (blue) are given with respect to the corresponding value of the all-aligned ones.

10. “By adding SOC, the band gaps of the 12- and 96-atoms cells converge to slightly different values - 0.45 153 eV and 0.54 eV. . . .”- is it a negative sign before 0.45, or a typo? The “-” symbol before 0.45 eV was supposed to be the beginning of a text insertion, which has to be closed after 0.54 eV with another “-” symbol, which is now missing in the text.

Thanks for pointing this out. The error has now been corrected in the revised manuscript.

11. In Table 2, the second value corresponds to octahedra tilt angles for 6188 atoms referring to pseudo-randomly oriented FA? If yes, the * is missing there.

Yes, the referee is right, it refers to the pseudo-randomly oriented FA. We thank the referee for catching this typo that we have now corrected.

12. Also in Table 2, the heading should be “Total Dipole PBE0 300 K (Debye/ABX3)”, and not “Total Dipole PBE0 3000 K (Debye/ABX3)”.

We thank the referee for catching this typo that we have now corrected.

13. “This again indicates the importance of the system size, in order to detect the possible formation of band gap domains.”—Is it possible to show some convergence with size of the cell in supporting information? Did the author try going beyond the 6144-atom cell? Would that give the same results?

We ran a small NPT AIMD trajectory at PBE level for a 20736-atoms supercell ($12 \times 12 \times 12$) for ~ 1 ps. As the referee can imagine, this is a very demanding simulation, especially considering that at least 5 ps are needed to equilibrate the system. Therefore, these results were not included in the manuscript. However these preliminary results indicate that band gap domains will reach convergence in real space for a system size of at least $12 \times 12 \times 12$, which is prohibitive for AIMD and would require classical molecular dynamics (CMD). CMD does not allow the calculation of electronic properties such as the band gap, but as shown in the manuscript, the band gap domains are strictly correlated with the octahedral tilting domains, which can be evaluated along the CMD trajectories. This is also supported by findings by the Walsh’ and Csányi’ groups (<https://doi.org/10.1002/sml.202303565> now included as a reference in the revised manuscript) on octahedral tilting patterns in CsPbI_3 where they show well defined octahedral tilting patterns extending over 8-12 primitive cells for a $20 \times 20 \times 20$ supercell.

14. Values reported in Table 3 are very confusing. It is not well explained that how the dipoles are calculated. In the third column, it says “Total dipole-FA dipole”, which does not match the values given in the column. Please explain how do you get the values.

We have added a paragraph to the “Methods” section to explain in detail how we compute the classical dipole moment. For the full discussion of how we compute the classical dipole

moment, see the response to the first referee’s comment 3. Since the dipole moment is a vector, in Table 3, we have reported the modulus of the total dipole moment of the entire system ($|\mathbf{d}_{Tot}|$), the modulus of the FA contribution to the total dipole moment ($|\mathbf{d}_{FA}|$), and the modulus of the difference between the total dipole moment vector and the FA contribution ($|\mathbf{d}_{Tot} - \mathbf{d}_{FA}|$). Since \mathbf{d}_{Tot} and \mathbf{d}_{FA} are vectors, $|\mathbf{d}_{Tot} - \mathbf{d}_{FA}|$ is generally different from $|\mathbf{d}_{Tot}| - |\mathbf{d}_{FA}|$. We understand that the column labels in the original manuscript could have been misleading. We have now replaced them with $|\mathbf{d}_{Tot}|$, $|\mathbf{d}_{FA}|$, and $|\mathbf{d}_{Tot} - \mathbf{d}_{FA}|$. We have also slightly revised the caption of Table 3.

Response to Referee #3

This manuscript presents a comprehensive computational investigation of size effects in α -FAPbI₃ perovskite using large-scale ab initio simulations. The authors employ both static DFT calculations and ab initio molecular dynamics to systematically examine how system size affects structural and electronic properties, with particular focus on band gap, structural distortions, and dipole moments. The authors present a systematic study employing multiple computational approaches and progressively larger system sizes, from 12 to 6144 atoms. Their analysis effectively establishes connections between structural features, electronic properties, and dipole moments.

We thank the referee for appreciating our multiple computational approach to study the structural and electronic properties of α -FAPbI₃.

However, several significant concerns need to be addressed. While the work is technically sound, similar conclusions about system size effects have been previously demonstrated for other hybrid perovskites, particularly MAPbI₃. The authors should more clearly articulate what new physical insights their study provides beyond confirming similar behavior in FAPbI₃.

The referee is right, the link between long-range structural correlations and optoelectronic properties in hybrid perovskites in general has already been pointed out, but to our knowledge no one has yet performed large-scale AIMD simulations to explicitly demonstrate this at the quantum mechanical level and at the nanoscale. In previous studies, classical molecular dynamics (CMD) complemented with experimental evidence has been successfully used to demonstrate the presence of nanoscale octahedral tilting domains (e.g. Weadock et al. DOI: 10.1016/j.joule.2023.03.01) that has now been included in the revised version of the

manuscript), but CMD simulations cannot provide direct information about the optoelectronic properties of the system. There are also studies where ab initio simulations on smaller systems have been combined to emulate a larger system using methods such as linear scaling three-dimensional fragment (Ma and Wang DOI: [dx.doi.org/10.1021/nl503494y](https://doi.org/10.1021/nl503494y)), which we had already cited in our original manuscript. Finally, there are several papers on polaritons and excitons in hybrid halide perovskites (e.g. DOI: [10.1021/acs.jpcclett.2c01945](https://doi.org/10.1021/acs.jpcclett.2c01945), DOI: [10.1126/sciadv.1701217](https://doi.org/10.1126/sciadv.1701217)) studied at the DFT level but on too small system sizes to get the right physics.

We have added the following reference to the revised version of the manuscript in the “Introduction” section when we discuss the literature on large scale simulations of hybrid halide perovskites:

- Weadock, N. J. et al. The nature of dynamic local order in $\text{CH}_3\text{NH}_3\text{PbI}_3$ and $\text{CH}_3\text{NH}_3\text{PbBr}_3$ Joule 7:5 1051-1066 DOI: <https://doi.org/10.1016/j.joule.2023.03.017> (2023).

More importantly, the PBE0+SOC calculations, which shown to be the most accurate approach, are only performed for systems up to 96 atoms. While the computational cost of such calculations is acknowledged, this limitation undermines the main conclusion that “the size of the simulated system needs to approach the nanoscale.” The authors should either extend these calculations to larger systems, provide evidence of convergence at 96 atoms, or explicitly discuss this limitation and its implications for their conclusions.

With Quantum ESPRESSO, it is impossible to go beyond 96 atoms due to the memory bottleneck, and unfortunately CP2K, which is much less memory demanding due to the mixing of plane-waves and localized Gaussian basis sets, has SOC implemented only up to 2D systems. This prevents us from performing convergence on PBE0+SOC as requested by the referee.

Following the referee suggestion, we have added this sentence in the conclusions of the revised version of the manuscript:

“A memory bottleneck prevented the completion of a convergence study on PBE0+SOC for the larger supercells; however, an assessment of the convergence on the smaller cells permitted the confident assumption of mutual cancellation of PBE0 and SOC effects also on the larger supercells. ”

The manuscript could be suitable for publication after addressing these concerns, particularly by adding explicit comparisons with previous MAPbI₃ studies and providing a more thorough analysis of PBE0+SOC convergence. The authors should also more explicitly discuss the limitations of using PBE results for larger systems.

The discussion and comparison with MAPbI₃ has now been expanded in the introduction to the revised version of the manuscript, as can be seen from the answers to the first question of the first referee and his/her own first question. As stated in the response to the previous question, explicit PBE0-SOC calculations on larger systems are currently unfeasible so that we have to rely on PBE values. The PBE limitations on larger systems are now explicitly addressed in the “Results” and “Conclusions” section by explicitly stating that even if the band gap is correct, the positions and symmetry of the valence band maximum (VBM) and conduction band minimum (CBM) obtained using the PBE method can differ significantly from their actual positions. A table reporting the computed energies for VBM and CBM is now included in the Supporting Materials of the revised version of the manuscript.

Response to referee comments

Virginia Carnevali,¹ Lorenzo Agosta,¹ Vladislav Slama,¹
Nikolaos Lempesis,¹ Andrea Vezzosi,¹ and Ursula Rothlisberger¹

¹*Laboratory of Computational Chemistry and Biochemistry,
Institute of Chemical Sciences and Engineering,
Swiss Federal Institute of Technology (EPFL), Lausanne, Switzerland*

Dear Editor,

Please find enclosed a revised version of our manuscript with the title “Nanoscale size effects in α -FAPbI₃ evinced by large-scale ab initio simulations”. The authors would like to thank the referees for their pertinent comments and suggestions, which significantly helped to improve the quality of this manuscript. In the following, we present detailed itemized responses/corrections to all the referee comments, first summarizing the main changes made in the revised manuscript. Here, all author responses are colored in blue while additions to the manuscript and Supporting Material are in green.

Response to Referee #1

The authors have revised the manuscript with new discussion and additional calculations. I have minor reservations, I wish the revised manuscript elaborated on how and why the size effect varies across these prototypical perovskites. A more explicit and in-depth analysis would enhance the novelty and rigor of the study beyond the standard expectations. After that, I have no further comments and the manuscript can be considered acceptable for publication.

We thank the referee for his appreciation of our work. Following the referee’s suggestion, we have added a new paragraph in the Introduction with a more explicit discussion of why the system size is a key factor in FAPbI₃ in comparison with the other prototypical perovskite structures. We have also added three new references.

Added paragraph:

In general, system size plays a crucial role in the accuracy and predictive power of DFT simulations of perovskite materials, its importance varying significantly for different classes

of perovskites. In organic-inorganic hybrid perovskites such as FAPbI₃ and MAPbI₃, the dynamic behavior of the organic A-site cations and the soft, anharmonic lattice framework introduce complex structural fluctuations and symmetry-breaking distortions that are not well captured by small simulation cells.^{39,41} The same is expected for the Sn counterparts, FASnI₃ and MASnI₃, but with a less pronounced system size effect due to the stiffer structures resulting from a more covalent Sn-I bond compared to the more ionic Pb-I bond.⁴⁵ For example, the orientation and rotation of the FA⁺ or MA⁺ cations affect hydrogen bonding with the halides, local electric fields, and ultimately the band structure, dielectric screening, and polaron formation.^{38,40} In contrast, inorganic halide perovskites such as CsPbI₃ and CsSnI₃ exhibit reduced orientational disorder due to the isotropic nature of the Cs⁺ ion, but still require moderately large supercells ($\sim 2 \times 2 \times 2$ or larger) to accommodate the octahedral tilting patterns and electron-phonon coupling effects that influence their optoelectronic behavior.⁴³ Meanwhile, oxide perovskites such as BaTiO₃ are relatively rigid and structurally less complex, allowing ferroelectric behavior and structural distortions to be reliably captured even in smaller supercells.^{46,47} However, even for oxide perovskites, larger cells may be needed to model phase transitions or strain effects.⁴⁸ Overall, while system size affects all perovskite simulations to some extent, the need for large, thermally sampled simulation cells is particularly pronounced in soft hybrid halide systems, where neglecting size and disorder effects leads to significant underestimation of band gap, carrier dynamics, and structural stability.

New references:

- Evarestov, R.A. Bandura, A.V. First-principles calculations on the four phases of BaTiO₃. *J. Comput. Chem.*, 33, 1123-1130. DOI: 10.1002/jcc.22942 (2012).
- Bayon, A., De la Calle, A., Ghose, K. K., Page, A., McNaughton, R. Experimental, computational and thermodynamic studies in perovskites metal oxides for thermochemical fuel production: A review, *International Journal of Hydrogen Energy*, 45, 23, 12653-12679, DOI: 10.1016/j.ijhydene.2020.02.126 (2020).
- Gigli, L., Veit, M., Kotiuga, M. et al. Thermodynamics and dielectric response of BaTiO₃ by data-driven modeling. *npj Comput Mater* 8, 209, DOI: 10.1038/s41524-022-00845-0 (2022).